

# Probabilistic downscaling of EURO-CORDEX precipitation data for the assessment of future areal precipitation extremes of different durations

Abbas El Hachem[1], Jochen Seidel[1], and András Bárdossy[1]

[1]Institute for Modelling Hydraulic and Environmental Systems, University of Stuttgart, D-70569 Stuttgart, Germany

**Correspondence:** Abbas El Hachem (abbas.el-hachem@iws.uni-stuttgart.de)

**Abstract.**

This work presents a methodology to inspect the changing statistical properties of precipitation extremes with climate change. Data from regional climate models for the European continent (EURO-CORDEX 11) were used. The use of climate model data requires first an inspection of the data and a correction of the biases of the meteorological model. Both the correction of biases of the point precipitation and those of the spatial structure were performed. For this purpose, a quantile-quantile transformation of the point precipitation and a spatial recorrelation method were used. Once bias-corrected, the data from the regional climate model were downscaled to a finer spatial scale using a stochastic method with equally probable outcomes. This enables the assessment of the corresponding uncertainties. The downscaled fields were used to derive area-depth-duration-frequency (ADDF) curves, and area-reduction-factors (ARF) for selected regions in Germany. The estimated curves were compared to those derived from a reference weather radar data set. While the corrected and downscaled data show good agreement with the observed reference data over all temporal and spatial scales, the future climate simulations indicate an increase in the estimated areal rainfall depth for future periods. Moreover, the future ARFs for short durations and large spatial scales increase compared to the reference value, while for longer durations the difference is smaller.

## 1 Introduction

Climate change became a very important issue in the past decades. It is likely to affect life conditions all over the world. One of the important issues in this context is how magnitudes and frequencies of extremes will change. A large number of studies were conducted to investigate this issue. In general, global climate models (GCMs) are used to model the past changes (forced by observations) in climate systems on a global scale and to project future changes (forced by emissions scenarios) (Randall et al., 2007). These are based on the numerical solution of mathematical representations of the physical processes and their interactions (such as the conservation of energy, mass, and momentum, the thermodynamic equations such as the gas law, and the connection of processes on land with the atmosphere and vice-versa) (Randall et al., 2007; Hennemuth et al., 2017). GCM provides output variables with a relatively coarse space-time resolution, typically between 100 and 500 $km^2$, and for 6-hour intervals (Stocker, 2014). However, the transfer and use of global results to local and regional climate analysis, especially for hydrological processes such as precipitation, requires a finer space-time resolution, which can be achieved



through downscaling. The latter can be divided into two categories: empirical-statistical downscaling (ESD) and dynamic downscaling via regional climate models (RCM) (Rummukainen, 2010). ESD exploits the statistical nonlinear relationships between small- and large-scale information about climate variables. RCMs use the GCM output as lateral boundary conditions and, coupled with parameterization schemes to account for local aspects (e.g., topography), climate data are acquired at a higher spatial resolution. Depending on the driving GCM model and the applied parameterization schemes, several RCM ensembles

are available (Kotlarski et al., 2014).Although GCM and RCM data provide essential information about climate systems, they cannot fully and correctly simulate all relevant spatial and temporal processes. Representative concentration pathway (RCP) provides information about possible future climate scenarios (Nakicenovic et al., 2000). The change in emission concentration is integrated into the GCM calculations and converted into carbon dioxide ($CO_2$) equivalents. An increase in the amount of greenhouse gases implies an increase in the global temperature and, hence, an alteration of the climatic system (Van Vuuren

et al., 2011; Pachauri et al., 2014). In the case of the Coordinated Downscaling Experiment for the European Domain (EURO-CORDEX) data, the governing GCM is driven by a set of several RCP scenarios (Nakicenovic et al., 2000). With RCP8.5 being the scenario with the largest increase in greenhouse gas emissions by the end of the 21st century. Due to the increase in temperature values, the maximum amount of water vapor in the atmosphere increases, which leads to an increase in the frequency and intensity of precipitation extremes (Li et al., 2021). However, the rate of increase is not exactly known, and the

change varies spatially and temporally (Singh, 2017).

Despite the ongoing advancement of the RCM data, their immediate applicability is fraught with challenges, including the existence of biases compared to observational data (e.g., frequency of occurrence of dry and wet values, precipitation intensity in extreme events, wet and dry spatial patches, systematic underestimation or overestimation), limitations in the spatial resolution (discrepancies in the model and observation spatial scale), and the correct representation of the spatial dependence

structures between the different locations. Kotlarski et al. (2014) found that precipitation values from several hourly RCM data, even at seasonal and regional scales, exhibit a bias of $\pm\ 40\ \%$ with a tendency to be overestimated. Meredith et al. (2021) examined the precipitation diurnal cycle for current and future periods of EURO-CORDEX. Most models exhibit timing errors in the occurrence of maximum hourly precipitation intensities. In all models, the peak occurred several hours before the one appearing in the observations. Several methods have been developed to reduce the bias in regional climate model (RCM)

simulations. Teutschbein and Seibert (2012) provided an overview of several bias correction methods, indicating that even simple approaches were able to mitigate the biases in the model data. Moreover, Maraun (2013) illustrated that bias correction methods based on quantile mapping of the cumulative distribution functions (CDF) of different spatial scales (e.g., points and grid cells) are a deterministic approach that results in a misrepresentation of the temporal and spatial variability. For reliable results, bias correction methods should be applied using reference data on the same spatial scale as the model data. However,

even the most sophisticated bias correction methods cannot handle large model errors and, if used incorrectly, can lead to false climate change signals (Maurer and Pierce, 2014; Maraun et al., 2017). To overcome these problems, approaches called trend-preserving have been developed, which aim to maintain the trend in the mean and in the higher quantiles (Hempel et al., 2013; Casanueva et al., 2020).



Lange (2019) and Volosciuk et al. (2017) suggested treating the bias correction and downscaling part separately. Bias ad-
justment should be performed at the same spatial scale between observations and the climate model output. Downscaling
climate model data to a finer spatial resolution, namely bridging the scale gap, should then be done using a stochastic rather
than a deterministic approach (e.g., interpolation) (Maraun, 2013). Furthermore, Widmann et al. (2019) compared the resulting
spatial variability of multiple downscaling methods and found that only the techniques that considered a multi-site behaviour
or directly modeled the spatial dependence gave a realistic representation of the spatial dependence structure. Bárdossy and
Pegram (2012) analyzed the spatial dependence of bias-corrected RCM daily precipitation values and showed how it is under-
estimated. The effects of the underestimation were clearly noticeable at larger spatial scales and led to an underestimation of
areal precipitation extremes. To address this, the authors introduced a recorrelation approach to correct the model dependence
structure. Switanek et al. (2022) implemented a stochastic downscaling scheme based on temporally and spatially corrected
RCM daily to transfer the coarse scale RCM data to a finer scale and derive spatially coherent daily precipitation time series.

Concerning the change in the frequency and intensity of precipitation extremes, an increase is projected especially for sub-
daily durations (Westra et al., 2014; Cannon and Innocenti, 2019; Fowler et al., 2021). However, Berg et al. (2019) examined
the derivation of summer depth-duration-frequency (DDF) statistics from hourly EURO-CORDEX data at the highest available
horizontal resolution (EUR-11) for several European countries (including Germany). National DDF reference curves were
used for comparison. Several RCMs were selected, and for the long duration the quality was considered fair, but for the short
duration the models showed poor representation of the hourly extreme values. The rainfall amount corresponding to a 10-year
return period was greatly underestimated by the RCMs' output. The problem lies in the way convection is represented. The
latter plays a major role in sub-grid processes and sub-daily rainfall extremes. A newer family of models, the convection-
permitting climate models (CPM), running at a finer horizontal resolution ($\leq 4$ km), show a better representation of the hourly
precipitation extremes (Meredith et al., 2021; Ban et al., 2021). However, additional development is required due to several
biases being present, and the high computational requirements limit their current large-scale applications (Kendon et al., 2014;
Berthou et al., 2020). Several approaches were proposed to derive future DDF curves (or adapt current DDF curves) from
RCM precipitation data. Martel et al. (2021) provided an overview of current methods. The first line of thinking consists of
modifying current DDF curves by a constant or a variable increase factor. The simple constant percentage increase applies a
constant increase factor (between 15 and 30%) to current DDF values. An alternative is the adaptive percentage increase that
utilizes an increase factor dependent on the projected temperature increase and rainfall frequency for future periods. The factor
changes for the different durations and return periods. Another approach is the percentage increase based on the Clausius-
Clapeyron relationship which relates the change in rainfall intensity to the local increase in temperature (per °C). All of the
aforementioned methods were based on upscaling of current DDF curves to future ones. Other methods, however, exist that
utilized the output of GCM and RCM. For example, Srivastav et al. (2014) derived future DDF curves using GCM data for
the region of Canada by an equidistant quantile mapping of the annual maxima. Spatial downscaling was used to transfer the
data to the point scale, and temporal downscaling was used to account for the changes between the historical and RCP future
projections. Mantegna et al. (2017) used data from a dynamical and high-resolution convection-parametrizing RCM to derive
sub-daily intensity duration frequency (IDF) curves. The latter were compared to two observation locations in Australia. The





future IDF curves suggest an increase in sub-daily rainfall intensities of 15% per °C. So et al. (2017) derived future IDF curves
for South Korea from daily RCM data through stochastic downscaling to sub-daily resolutions incorporated into a Bayesian
inference framework. The results indicate an increase in the expected rainfall from 5 to 30% under the RCP8.5 scenario. Other
works also exist, for example, Hosseinzadehtalaei et al. (2018) derived future IDF for Belgium, Forestieri et al. (2018) for
Sicily in Italy and for Iran (Khazaei, 2021).

Many of the previous studies consider deriving the DDF (or IDF) for the point scale. Traditionally area reduction factors
(ARF) are used to transfer the point value to the areal (or catchment) scale. These are in general based on simple assumptions,
and the effect of climate change is not considered. However, the consequences of heavy precipitation, such as flooding, are
related to the volume of water, so the spatial aspect should not be ignored. This work considers precipitation as a spatial
phenomenon, without purely point statistics, and aims to assess the expected change in future areal precipitation extremes
under the RCP8.5 scenario compared to reference data and the present period. This is done by calculating area-depth-duration-
frequency curves (ADDF) over several durations (hourly to daily) and over different spatial scales from 1 to 1000 $km^2$.
Essential questions to be investigated in this work are:

1. Knowing that precipitation extremes look different depending on scales, to what extent can the climate model produce
   extremes correctly?

2. How can spatial and temporal dependence structures of climate models be successfully corrected?

3. How will the statistics of areal extremes change with climate projections?

To answer these questions, the following scheme was undertaken. The first step consisted of upscaling the point data to
the model scale. This enabled deriving a reference for temporal and spatial dependence structures. A recorrelation procedure
was implemented to correct the model spatial dependence structure to match the reference. In the second step, the bias in the
magnitudes of future projections and any subsequent bias in the marginal distribution function was corrected using a double-QQ
transformation. The corrected data on the model scale were then spatially downscaled using a conditional simulation method
(Bárdossy and Hörning, 2016b, a). From the downscaled spatial fields, areal precipitation statistics were derived and compared
to those extracted from weather radar data. The downscaling and analysis of areal precipitation extremes are showcased in the
third section of this manuscript. The results are discussed and a conclusion involving key messages finalizes the paper.

## 2  Study area and data description

The study area is delimited by the weather radar region of Hannover located in the state of Lower Saxony in Germany. The
weather radar is positioned at the Hannover airport and has a coverage radius of 128 km. The north of the region is generally
flat, and moving towards the southeast the elevation increases to 1141 m a.s.l. in the Harz Mountains. The average yearly
precipitation ranges from 500 to 1700 $mm\,yr^1$ (Haberlandt and Berndt, 2016b). Following previous studies on the space-time
statistics of rainfall extremes, the latter was selected as the study area (El Hachem et al., 2022; Goshtsasbpour et al., 2022).





Within this region, the German Weather Service (DWD) operates a precipitation measurement network of 127 rain gauges with sub-hourly resolution (DWD Climate Data Center (CDC), 2021a). The data for these stations were acquired for the years between 2005 and 2020. Moreover, the hourly weather radar data (RADOLAN-RW) for the period 2005-2020 were used. The latter is the operational DWD radar composite. It is a merged product of the derived precipitation fields from all available weather radars and the DWD rain gauges. The data were made available by the DWD (DWD Climate Data Center (CDC),

2021b). For this study, only the observations lying within the radar area of Hannover were extracted. The third data set is the EURO-CORDEX data. These have been provided within the Coordinated Regional Downscaling Experiment (CORDEX) for the European continent with two horizontal simulation domains of 50 km (EUR-44) and 12.5 km (EUR-11). The simulation output consists of several data sets with hourly resolution representing different atmospheric and surface-near variables such as precipitation (Jacob et al., 2014). The REgional MOdel (REMO) is especially advantageous for precipitation analysis on the

hourly scale since advection is integrated within the parameterization scheme (Jacob and Podzun, 1997; Jacob, 2001). In this work, the MPI-M-MPI-ESM-LR-GERICS-REMO2015-v1 developed by the Max-Planck-Institute für Meteorologie (MPI-M) and the Climate Service Center Germany (GERICS) was used. The data were made available by the ClimXtreme Central Evaluation System framework (Kadow et al., 2021).

Figure 1 presents the locations of the EURO-CORDEX 11 ° center grid points in Germany and in the radar area of Hannover

(black circle). To avoid edge effects in the interpolated and simulated fields, only model points falling within a 10-kilometre inward buffer (red circle) of the radar coverage boundary were selected. The DWD rain gauges are visualized as orange triangles. The red points in the lower right map present the center of the radar pixels. The weather radar grid with a spatial resolution of 1 kilometre constitutes the interpolation and simulation grid. The orange box is a 12.5*12.5 km polygon presenting one EURO-CORDEX grid cell (centered around the green point). In total 273 grid cells are available within the selected area.

**Figure 1.** Map of Germany with the EURO-CODEX 11 ° grid center locations. The study area defined by the weather radar area of Hannover along the DWD rain gauge data and the radar grid is shown. The background map is credited to © Google Maps.

The goal is to find the reliability/usability of the RCM model data for areal precipitation analysis with a special focus on extremes. Since the model data consist of grid cell averages, it is not suitable to compare them to rain gauge (point) data. Hence, the first reference data was based on interpolated fields using the DWD station data (refered to as $DWD_{point}$). The interpolation was carried out using Ordinary Kriging and the weather radar observation grid of 1 km was used as the interpolation grid. The resulting fields were then spatially aggregated to match the EURO-CORDEX 11° grid. This reference data set will be denoted hereafter as $DWD_{interp}$. The second data set is the spatially averaged radar data for the period 2005-2020. The aggregated data match the model grid and are denoted hereafter as $Radar_{avg}$. Note that in all cases the arithmetic mean of the 1 km pixels falling within each model cell was calculated and assigned to the corresponding model pixel. A summary of the used data sets is presented in Table 1.





**Table 1.** Description of the used data sets along their availability and corresponding spatial and temporal resolutions.

| Notation | Data description | Period | Spatial resolution | Temporal resolution |
|---|---|---|---|---|
| $\text{DWD}_{point}$ | Rain gauges | 2005-2020 | Point scale | Hourly |
| RADOLAN-RW | Weather radar data | 2005-2020 | 1 km | Hourly |
| $\text{DWD}_{interp}$ | Interpolated fields | 2005-2020 | 12.5*12.5 km | Hourly |
| $\text{Radar}_{avg}$ | Radar fields | 2005-2020 | 12.5*12.5 km | Hourly |
| $\text{EURO-CORDEX}_{hist}$ | Historical data | 1970-2005 | 12.5*12.5 km | Hourly |
| $\text{EURO-CORDEX}_{rcp}$ | RCP data | 2006-2099 | 12.5*12.5 km | Hourly |

## 3 Methodology

The procedure for analyzing, correcting, and downscaling the RCM data is divided into three parts and is presented in Figure 2. First, the spatial dependence structure of the model needs to be corrected according to a reference-based structure. The correction of the distribution of the point precipitation amounts is done subsequently. In order that the biased precipitation amount distribution should not influence the spatial structure, it was derived on the basis of the pixel-wise rank correlation using a reference data set (for example, $\text{DWD}_{interp}$). The correction is based on a recorrelation procedure described in section 3.1. The dependence structure affects greatly the areal extremes, especially for large spatial events (Bárdossy and Pegram, 2012). This part refers to section (a) of the flowchart in Figure 2. Once the dependence structure has been corrected, any remaining bias in the marginal distribution function of future climate projections has to be handled. For this, a double-QQ transformation involving information from the reference data and the model historical data was applied (Bárdossy and Pegram, 2011). This section represents part (b) of the flowchart in Figure 2. Afterwards, the final corrected data are downscaled using a stochastic simulation algorithm (random mixing, (Bárdossy and Hörning, 2016b)) to the finer spatial resolution of 1 km. The final fields are eventually used for the analysis of the spatial extent of extremes via area-depth-duration-frequency (ADDF) curves. The ADDF curves were calculated on the pixel-scale (1 $km^2$ area), for different areas (up to 1024 $km^2$) and durations (hourly to daily). The possible impact of climate change on the statistics of areal extremes was to be investigated. Note that this section refers to part (c) of the flowchart in Figure 2





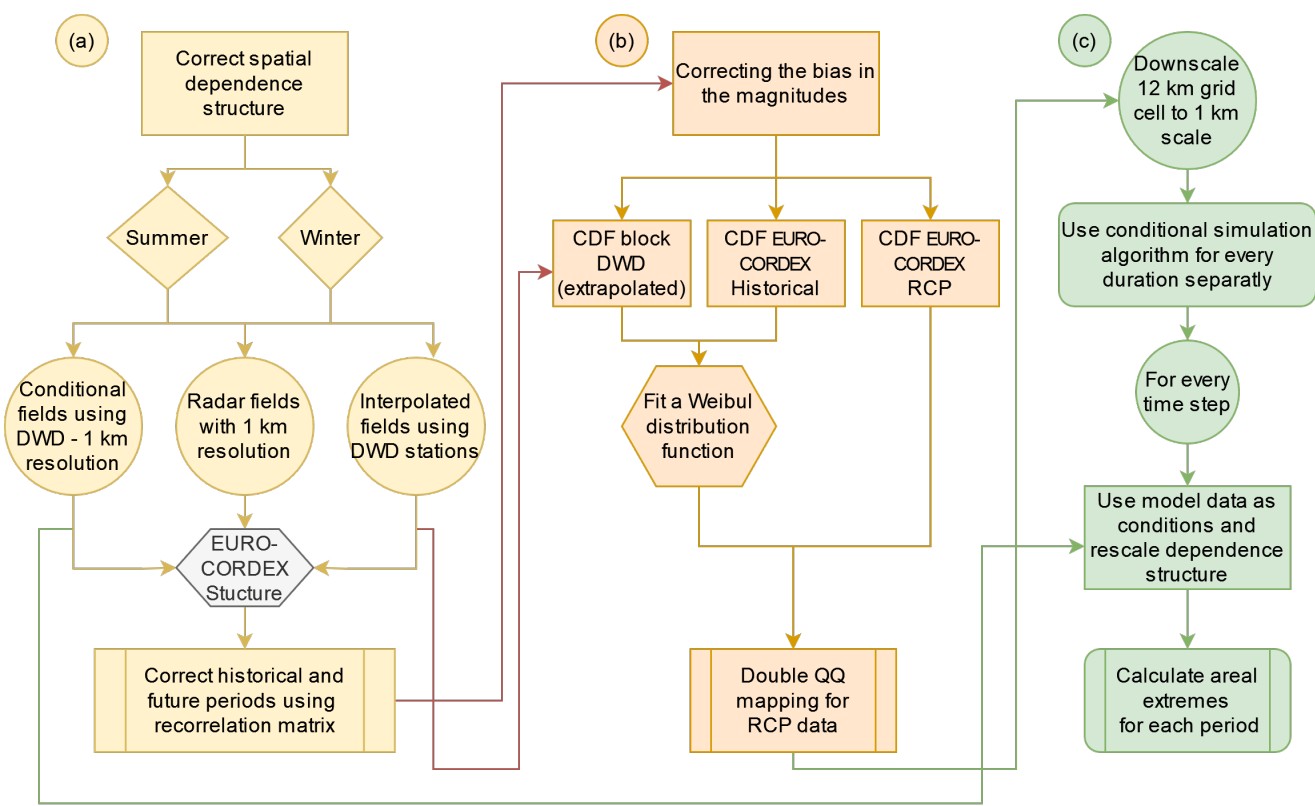

**Figure 2.** Flowchart describing the methodology for correcting the spatial (part (a)) and temporal (part (b)) structures of EURO-CORDEX 11 ° data. The corrected values are then downscaled (part (c)) and used for analysis of areal extremes.

## 3.1 Correction of dependence structure

The spatial dependence of precipitation plays a major role in the distribution of areal rainfall and the corresponding extremes. An often neglected problem is whether regional climate models can replicate the observed dependence structure. In (Bárdossy and Pegram, 2012) this problem was discussed and a possible solution was presented. A similar procedure was here implemented, however, using a finer spatial and temporal resolution and instead of the Pearson correlation a rank correlation type method was used with normal score transformed variables. To account for different precipitation mechanisms and characteristics, the reference and model data were divided between the summer (April-September) and winter (October-March) seasons, and for each period the dependence structure was derived. A special challenge while working with sub-daily and especially hourly precipitation data is the large number of 0 mm precipitation values (around 80%).

Figure 3 displays the pair-wise normal score transformed rank correlation values for each data set individually plotted against the separating distance between each and all other grid cells. For the EURO-CORDEX data, the historical and future periods present similar behavior and only the historical data are displayed. Note that each correlation matrix has a size of 273 by 273. Panel (a) shows the correlation structure derived from the DWD rain gauges $DWD_{point}$ (black dots) and the spatially averaged





DWD$_{interp}$ data (red dots). The DWD$_{point}$ correlation values are lower than those of the DWD$_{interp}$ data. This difference can be justified theoretically - assuming stationarity of the spatial dependence - as the interblock variability leads to the reduction

of the variance. In fact, the covariance between two grid cells $V_i$ and $V_j$ can be written as a function of the covariance function of the point values $C(x,y)$:

$$\text{Cov}(V_i, V_j) = \frac{1}{|V_i|} \frac{1}{|V_j|} \int\limits_{V_i} \int\limits_{V_j} C(x,y) \, dx \, dy \tag{1}$$

While for the variance of the grid cell $V$:

$$\text{Var}(V) = \text{Cov}(V,V) = \frac{1}{|V||V|} \int\limits_{V} \int\limits_{V} C(x,y) \, dx \, dy \tag{2}$$

Both the covariance and the variance decrease. The decrease of the covariance is less than or equal to that of the variance, thus the correlation increases.

Thus due to scale difference, using the DWD$_{point}$ correlation structure as a reference for the correction of the EURO-CORDEX spatial structure would be incorrect.

In panels (b) and (c), the red dots represent the DWD$_{interp}$, the blue dots the Radar$_{avg}$, and the orange dots the EURO-

CORDEX data. Panel (b) for the winter period and panel (c) for the summer period. The results for DWD$_{interp}$ show a typical behavior of decreasing correlation with increasing separating distance associated with a large scatter. In both cases, the correlation structures of Radar$_{avg}$ present a smaller scatter and fall below those of DWD$_{interp}$ and Euro-Cordex. In other words, Radar$_{avg}$ shows less spatial continuity (a quick drop of correlation) and larger variability between the grid cells. Compared to DWD$_{interp}$ the EURO-CORDEX results show an underestimation of the dependence structure, especially in the summer

period. The method aimed to adjust the model dependence structure to match the reference over all temporal aggregations.





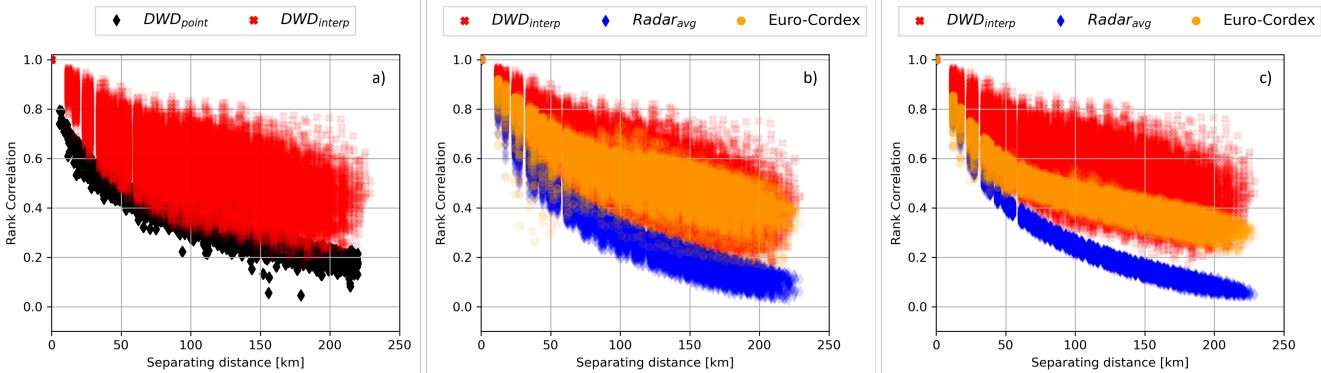

**Figure 3.** Panel (a) shows the rank correlation values between the DWD rain gauges (DWD$_{point}$) for the winter period (black dots) and between the DWD$_{interp}$ grid cell values (red dots) for the same period. Panels (b) and (c) show the calculated grid cell pair-wise rank correlation values from interpolated fields (red points), radar fields (blue points), and EURO-CORDEX fields (orange dots) for the winter and summer periods, respectively. The x-axis refers to the separating distance between the grid cells. Note that the white vertical spaces refer to the neighboring grid cells separating distance of $\approx$ 12.5 km.

The procedure presented by Bárdossy and Pegram (2012) used a mixed-type distribution, defined by a censored Gaussian copula to transform the daily precipitation data to the normal space, and a matrix recorrelation procedure based on linear algebra was implemented. The aim was to recorrelate the model data to obtain the same reference dependence structure. The latter was derived by calculating the Pearson correlation between the daily precipitation time series at the different grid cell

locations. A similar procedure was implemented in this section, however, using the theoretical correlation of the normal score transformed data.

A normal score transformation cannot be performed directly due to the large portion of zero values. Thus, instead, indicator series were used. The indicator series $I(t, u)$ were calculated for the reference and model data given a probability level $\alpha$. Several threshold values were tested, and the value $\alpha = 0.9$ was selected as it provided the best recorrelation results.

Let $F(t, u)$ be the distribution function of the precipitation time series $Z$ at location $u$. The indicator series can be calculated using equation 3. After converting the reference and model data to indicator series (0 and 1), the pair-wise correlation between two indicator series $\rho_i(u, v)$ was calculated. The indicator correlation matrix for the reference data is defined by $R_i$ and for the model data by $M_i$.

$$I(t, u) = \begin{cases} 1 & \text{if } F(Z(t, u)) > \alpha \\ 0 & \text{else} \end{cases} \tag{3}$$

For a given probability value $\alpha$, there is a one to one relationship between the indicator correlation corresponding to two locations $u$ and $v$ ($\rho_i(u, v)$) and the correlation of a bivariate Gaussian variable ($\rho_g(u, v)$) which has the same indicator correlation. Equation 4 describes this relation quantitatively.





$$\rho_g(u,v) = \frac{1}{2\pi\alpha(1-\alpha)} \int\limits_{0}^{arcsin\,\rho_i(u,v)} exp\Big(\frac{-y_\alpha^2}{1+sin\,t}\Big)\,dt \qquad (4)$$

Here $y_\alpha$ corresponds to the $\alpha$ probability standard normal distribution function $y_\alpha = \Phi^{-1}(\alpha)$.

The estimation of $\rho_g(u,v)$ has to be carried out numerically. Using the relation in equation 4, both indicator correlation matrices (describing the correlation between the grid points) of the reference and model data $R_i$ and $M_i$ were transformed to Gaussian correlation matrices $R_g$ and $M_g$. To ensure that the correlation matrices for the grids, $R_g$ and $M_g$, are positive semi-definite, minor modifications to the correlation values were undertaken while minimizing the distance between the original matrices ($R_g$ and $M_g$) and the modified matrices ($R_g^*$ and $M_g^*$) (Bárdossy and Plate, 1992).

From this point on, the matrix $R_g^*$ was transformed to the matrix $S_g$ by calculating the square root of applying single value decomposition (SVD) to $R_g^*$. Afterward, the correlation matrix $M_g^*$ was transformed to $T_g$ by calculating the inverse square root of applying SVD to $M_g^*$. The recorrelation matrix $F = T_g S_g$ was used to recorrelate $M_g^*$ to $R_g^*$ by matrix multiplication $V_g = M_g^* F$. The values in $V_g$ are in the Gaussian space and can be back-transformed to indicator correlations using a QQ transformation. Eventually, the model precipitation data were recorrelated using the recorrelation matrix $F$. Any negative

values after recorrelation were set to 0 precipitation value. The flowchart in Figure 4 gives an overview of the recorrelation steps.

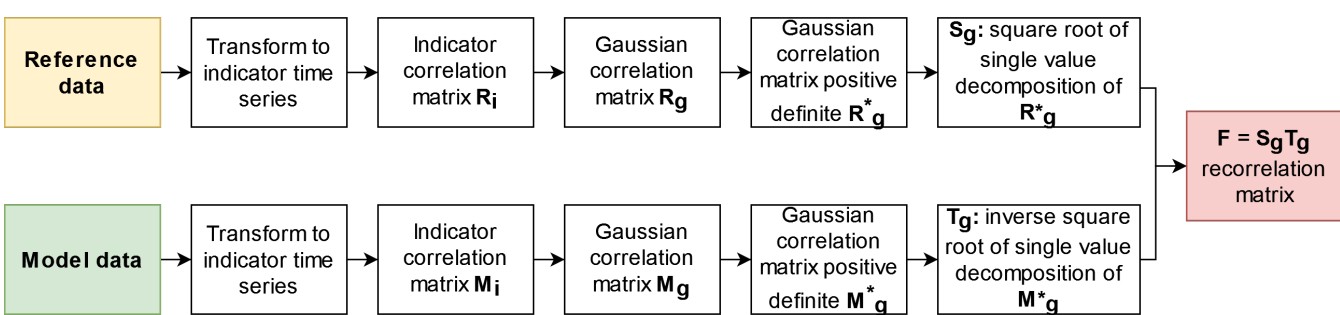

**Figure 4.** Flowchart describing the methodology for recorrelating the EURO-CORDEX data to have a similar dependence structure as the reference data.

The correction procedure should correct the dependence structure of the modeled precipitation. As dependence can be measured with different statistics a set of different possibilities is shown in Figure 5. In all panels of Figure 5, the x-axis and the y-axis values refer to the reference and model data correlation values, respectively. In panel (a), the blue points represent the

indicator correlation values, namely the matrices $R_i$ and $M_i$. In panel (b), the red dots are the correlation values after the normal score transformation and the green and blue ones after the correction of the spatial structure. The Pearson and rank correlations between the reference and model data were calculated and are presented in panels (c) and (d), respectively. In panel (c), the original Pearson correlation values (black points) showcase an improvement after the correlation procedure (orange points).





Meanwhile, in panel (d), the rank correlation values show an improvement after the correction (points in green). Similar results

were obtained for the winter period.

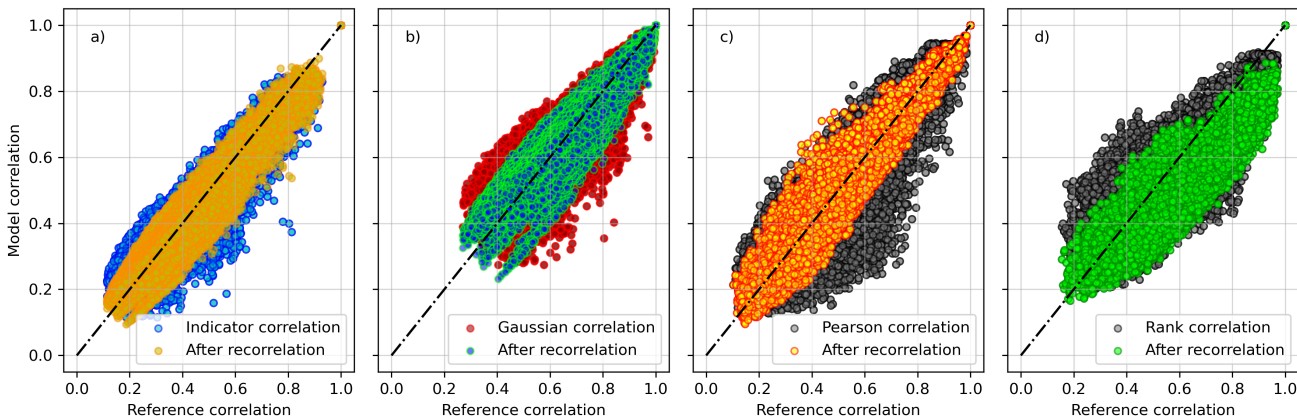

**Figure 5.** Panel (a) shows the indicator correlation values between the reference $DWD_{intep}$ (x-axis) and the EURO-CORDEX data (y-axis) before correction (blue dots) and after correction (yellow dots). In panel (b), the Gaussian correlation values are displayed before correction (red dots) and after recorrelation (green-blue dots). In panel (c), the Pearson correlation values before (black dots) and after recorrelation (orange dots) are shown. In panel (d) the rank correlation values before (black dots) and after recorrelation (green dots) are displayed. The reference and model dependence structures were calculated for the summer period.

The recorrelation matrix $F$ derived from the historical observations for each season separately can be used to recorrelate projected RCP scenarios. This is possible since the model's historical and future dependence structures are highly similar (see Figure A1). In Figure 6, the correlation values of the recorrelated grid cells are calculated as a function of the separating distance. Similar to Figure 3, the values for $DWD_{interp}$ (red dots), $Radar_{avg}$ (blue dots), and the original EURO-CORDEX

historical data (orange dots) are displayed. The results indicate that recorrelation procedure improved the model dependence structure compared to the original data for both seasons.





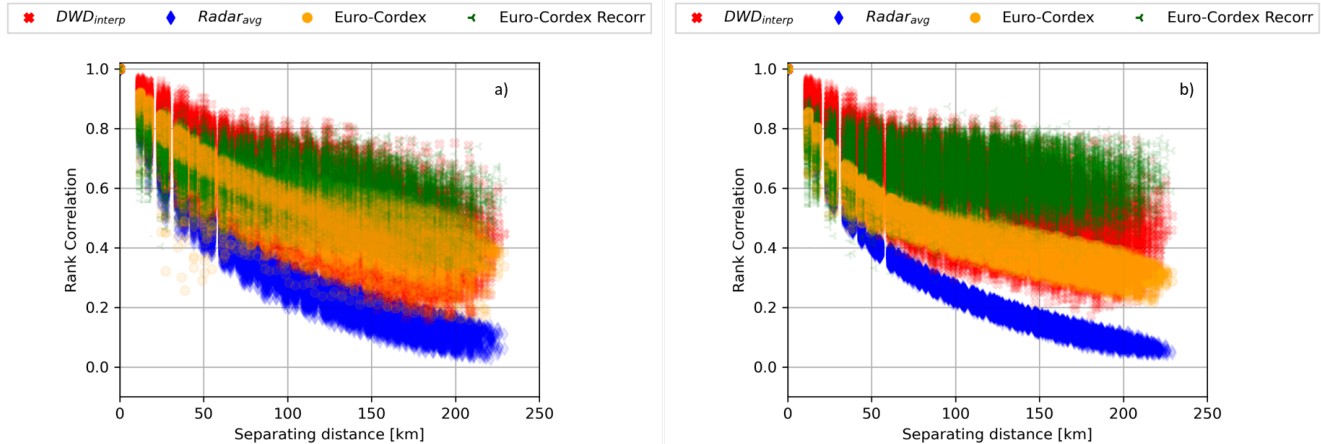

**Figure 6.** Panels (a) and (b) show the calculated grid cell pair-wise rank correlation values for the winter and summer periods from $\text{DWD}_{interp}$ (red points), $\text{Radar}_{avg}$ (blue points), and EURO-CORDEX fields (orange dots). The dots in green refer to the rank correlation of the EURO-CORDEX fields after recorrelation. The x-axis refers to the separating distance between the grid cells.

The recorrealtion procedure was done for every duration (from hourly to daily) separately. This provided consistent results for all durations. Note that this complete part refers to section (a) of the flowchart in Figure 2. After correcting the dependence structure, the marginal distribution function of the model's future data were corrected using a double QQ transformation.

## 3.2 Double quantile-quantile mapping

In statistics, quantile-quantile plots (QQ plots) are used to compare two distributions and identify if they both belong to the same distribution function. Often a test distribution is compared to a theoretical one. The comparison is based on the quantiles. Namely, a scatter plot between the quantiles of both data values is constructed. If the data have the same distribution function, the QQ plot will be defined by a linear function (y=x). To derive the QQ plot, the data of the two distribution functions are sorted, and their quantiles are calculated. In this section, a QQ mapping was applied to correct the bias in the projected model data while preserving the ranks of the values. An example of this was shown by Bárdossy and Pegram (2011) where the distribution function of regional climate models (RCM) was corrected using a double QQ transformation as defined by equation 5. For this, the CDF of the upscaled reference data at location X (for example $\text{DWD}_{interp}$) and the CDF of the recorrelated model data for the historical and future periods for the same location X were used. For the observation and historical data, a Weibull distribution function with three parameters was fitted using the maximum likelihood method (Singh, 1987). The latter was chosen as it provided the best fit.

$$Z(x,t) = F_o^{-1}(F_R(Z_R(x,t),x),x)$$

(5)

Where:





| $x =$ | target location |
| --- | --- |
| $t =$ | time step |
| $Z(x,t) =$ | corrected precipitation value |
| $F_o^{-1} =$ | inverse of the fitted CDF to the reference data |
| $F_R =$ | CDF of the RCM data |
| $Z_R(x,t) =$ | precipitation simulated by the RCM |

Due to the fact that the rain gauges might have missed some of the precipitation extremes, the fitted distribution function to $\text{DWD}_{interp}$ was extrapolated (for $Z > Z_m$) using an exponential distribution function with a single parameter $\lambda$ (Yan and Bárdossy, 2019). The extrapolation using the exponential distribution is defined by equation 6. Let $(Z_m, U_m)$ define the pair of the largest precipitation observation and corresponding quantile, the parameter $\lambda$ is calculated following equation 7.

$$F(z) = \begin{cases} 1 - \exp^{-\lambda z} & \text{if } z > 0 \\ 0 & \text{else} \end{cases} \tag{6}$$

$$\lambda = -\frac{1}{Z_m} \ln(1 - U_m) \tag{7}$$

    This enables correcting the RCP future data while allowing for maximum values to exceed the current observed values.

    In panel (a) of Figure 7 the x-axis refers to precipitation in millimeters and the y-axis to the cumulative probabilities. The red curve is the distribution function of the reference data $\text{DWD}_{interp}$. The green curve is the RCM distribution for the historical period and the blue curve shows the RCM curve for a future scenario. For every precipitation value in the future data, the
corresponding value in the historical data is found and for that, the corresponding value for the same quantile level in the observations data is assigned as the future value.

    An example of this is presented in Figure 7. In panel (a), the blue curve refers to the EURO-CORDEX historical data after the recorrelation procedure, the green curve to the recorrelated RCP8.5 data, and the red curve to the reference data $\text{DWD}_{interp}$ for the observation period. The black curve shows the double-QQ corrected RCP8.5 distribution function using the previously
described procedure. Panel (b) displays a scatter plot of the values after recorrelation (gray dots) and after the recorrelation and double-QQ mapping (blue dots). The red line refers to the raw RCP8.5 data. Panel (c) displays the CDF of $\text{DWD}_{interp}$, $\text{Radar}_{avg}$, the EURO-CORDEX historical, and the RCP8.5 data before and after correction for the same grid cell location. The $\text{Radar}_{avg}$ indicate a smaller maximum value compared to $\text{DWD}_{interp}$, though both distributions show few differences. Similarly to the recorrelation procedure, the double-QQ correction was done for every duration separately.
Note that this section refers to part (b) in the flowchart in Figure 2. This double-QQ transformation reduces the bias in the model data while preserving the signal in the RCP data (Bárdossy and Pegram, 2011). The final recorrelated and double-QQ corrected data are used for downscaling and investigating the areal rainfall statistics.





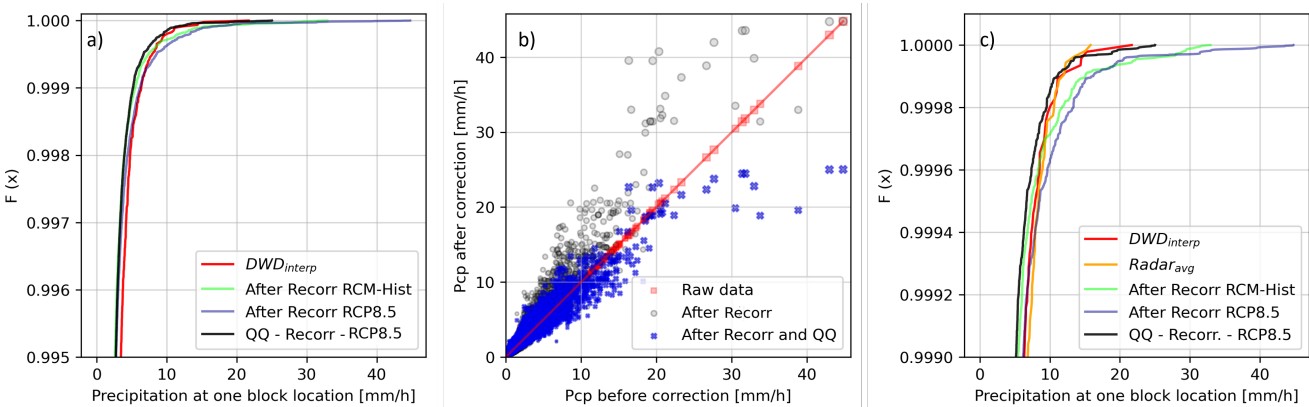

**Figure 7.** Panel (a) shows the upper 0.5% of the CDFs of the $DWD_{interp}$ (red curve), the EURO-CORDEX historical data (green curve), the recorrelated RCP8.5 data (blue curve), and the double QQ-corrected RCP8.5 data (black curve). Panel (b) shows the scatter of the values before and after the recorrelation (points in gray) and the double-QQ mapping (points in blue) for one example grid cell location. Panel (c) displays the upper 0.1% of the CDFs along the $Radar_{avg}$ values (orange curve).

## 3.3 Depth-duration-frequency (DDF) curves

To estimate rainfall depth for design values, a statistical analysis of rainfall maxima derived from long records is required.
In general, design values are associated with the corresponding duration and return period. The previous concept forms the basis of statistical analysis of heavy rainfall. Rainfall maxima are extracted from the observation time series for the different durations either by considering the yearly maxima (annual series) or the values exceeding a minimal threshold (partial series). The relation between rainfall depth, duration, and return period is commonly known as the Depth-Duration-Frequency (DDF) or Intensity-Duration-Frequency (IDF) curve. The idea behind the DDF curve is to derive a mathematical expression relating
the average rainfall intensity (i) occurring over a timescale (d) for a predefined return period (T) (Koutsoyiannis and Papalexiou, 2017). DDF curves are used to estimate the probability of non-exceedance of a certain rainfall amount for a given duration. These can be derived by a frequency analysis of the observed station data for different durations. Standard practice is to fit to the empirically calculated DDF curve a theoretical extreme value distribution function (e.g., Gumbel Type I), from which one can derive the possible rainfall depth (or intensity) for a certain return period and timescale. The reasoning behind fitting a
distribution function to the sampled annual or partial maxima is that these represent only one realization of the possible rainfall values for the corresponding duration.

In this work, the ADDF curves were derived from the observed annual series of 15 years (2005-2020), following the procedure described by the German Association for Water, Wastewater and Waste in DWA-A 532 (DWA-A, 2012). The extreme value distribution type I, known also as the Gumbel distribution function, is applied in the following form:

$$h_N(T_n) = u_j + w_j(-ln\ ln\frac{T_n}{T_n-1})$$ (8)





Where:

$$h_N = \qquad \text{Rainfall depth in [mm]}$$
$$T_n = \qquad \text{return period of the annual maxima in years [a]}$$
$$u_j, w_j = \qquad \text{parameters of the distribution function}$$

### 3.4 Area Depth-duration-frequency (ADDF) curves

For precipitation volumes, the consideration of the areal mean is required. The calculation of the Area-DDF (ADDF) curves is
not straightforward, as in contrast to point precipitation areal precipitation is not measured but has to be estimated. Further, the
distribution of the areal precipitation amounts differs from the distribution of point values.

For example, using the weather radar data for a target location of an area of 1000 $km^2$, all pixels located within the area
are identified, and for every time step the average of all the pixels is calculated. Eventually, a time series is acquired and used
as input to the DDF calculation procedure. Bennett et al. (2016) first suggested the use of interpolated rainfall data for the
direct estimation of the statistics of areal extremes by introducing the intensity-duration-frequency-area curves. The need for
ARF to convert points to spatial rainfall would be then eliminated. Radar-derived precipitation data can be used to calculate
the ADDF curves. However, the quality of the radar reference data and the relatively short observation time period highly
influence the results (Haberlandt and Berndt, 2016a). Marra and Morin (2015) used radar QPE to derive IDF curves for the
region of Israel and compared the results to nearby rain gauges. Despite efforts to reduce errors in the radar QPE, the final
results showed an overestimation of radar-derived IDF curves. An effect that increased with larger durations. Another study
done by Ghebreyesus and Sharif (2021) using the radar QPE data over the state of Texas to derive state-wide IDF curves
showed mostly an underestimation of short-duration maxima. The goal of using the radar QPE data is to be able to derive
spatially and temporally reliable IDF curves, eliminating the need for area reduction factors (Ghebreyesus and Sharif, 2021).

Areal precipitation extremes derived from weather radar data are prone to errors (Schleiss et al., 2020). For instance, due
to attenuation, beam blockage, sub-scale variability, and simplified Z-R relation (Villarini and Krajewski, 2010). However,
as there is still no possibility to correctly measure areal extremes, the error term cannot be easily quantified. In addition, the
rain gauge observations are sparse. Many short-duration intense rainfall events cannot be correctly sampled, and many are
completely missed (Lengfeld et al., 2020). The high spatial and temporal availability of the RADOLAN-RW merged station-
radar data makes it advantageous for the analysis of areal extremes and was used in this study for the estimation of reference
ADDF curves (DWD Climate Data Center (CDC), 2021b). However, due to the limited available temporal period, the ADDF
curves were derived for return periods of 5 years. Such short return periods are relevant for the design of urban drainage
networks.

### 3.5 Downscaling model to point scale

In order to better assess the quality of the EURO-CORDEX model data when representing areal precipitation extremes, a
direct comparison between point observations and model values is not reliable due to scale differences. Therefore, a spatial
downscaling of the model grid cell values to a finer scale is required. A deterministic approach, such as an interpolation





technique, can only provide one possible subscale realization without any estimate of an uncertainty interval. In general, most interpolation techniques can only provide smoothed fields because, in all Kriging applications, the equation system is solved by minimizing the estimation variance. Hence, the interpolated fields have less variability than the original one. However, for

hydrological applications, many processes are driven by variability rather than the average values.

To this end, a simulation of many possible realistic realizations with the same dependence structure can offer the corresponding variability. In general, there are two main types of simulation methods: conditional and unconditional simulations. Random mixing is a conditional simulation method that allows the stochastic generation of realizations that satisfy multiple conditions (Bárdossy and Hörning, 2016b). The method is extended from the work of Hu (2000) regarding the gradual deformation of

Gaussian fields. In order to achieve the goal of having realistic realizations, the following criteria must be fulfilled by the simulated field:

1. Match the measured data (conditioned on observations).

2. Have the same spatial dependence structure (represented by a variogram or spatial copula function).

3. The realizations should have a similar range as the observed values (no extreme values).

The random mixing methods allow capturing the spatial dependence structure through a spatial copula function. A copula is a mathematical function used to derive and model the dependence between variables independently of their distribution functions. If a spatial copula function was used, the asymmetry of the spatial field could be better taken into account (asymmetry can be viewed as the measure of skewness of univariate data). Another advantage of random mixing is the possibility to include multiple conditional observations that can be integrated as linear (equality) or non-linear (inequality) constraints. As

a stochastic method, several equally possible realistic realizations of a certain event (or time step) can be derived along the associated mean uncertainty field.

In this work, random mixing was used to upscale and downscale the $\text{DWD}_{point}$ observations to the EURO-CORDEX scale and vice-versa. The RMWSPy Python package was used to that end (Hörning and Haese, 2021). First, based on the $\text{DWD}_{point}$ data for the period 2005-2020, conditional fields were simulated for every time step with precipitation $> 1$ mm. The simu-

lated fields were conditioned on the $\text{DWD}_{point}$ observations, the corresponding marginal distribution function, and the fitted covariance (or variogram) model representing the spatial dependence structure.

In this step, the calculated experimental variogram was normed and saved. Note that because the variogram was calculated in the rank space, its estimation was smoother than by using the original observations. Eventually, a K-mean clustering was applied and three clusters of variograms were identified (Hartigan and Wong, 1979). Different numbers of clusters were tested,

but the chosen number of clusters (3) was found to be the most suitable as the fitted models were the most distinct. For each cluster, the mean variogram was calculated and an exponential variogram without a nugget was fitted. Since the EURO-CORDEX data are on the grid cell scale, the estimated spatial model is smoother than that of the point scale. Hence, for downscaling, the spatial dependence model calculated from the grid cell values needed to be rescaled to the point scale. To that end, using the EURO-CORDEX data the experimental variograms were calculated, and a similar clustering approach was





applied. This allowed assigning for every time step (for every duration) a suitable variogram. A similar approach of variogram clustering was presented in Bárdossy et al. (2021).

Figure 8 shows the fitted exponential variograms to each average cluster variogram from the point and model data. The grid cell variograms show similar behaviour as those derived from the point scale but with a smaller variance and a larger range. In other terms, for the same separating distance, the variance of the grid cell model is smaller than that of the point model.

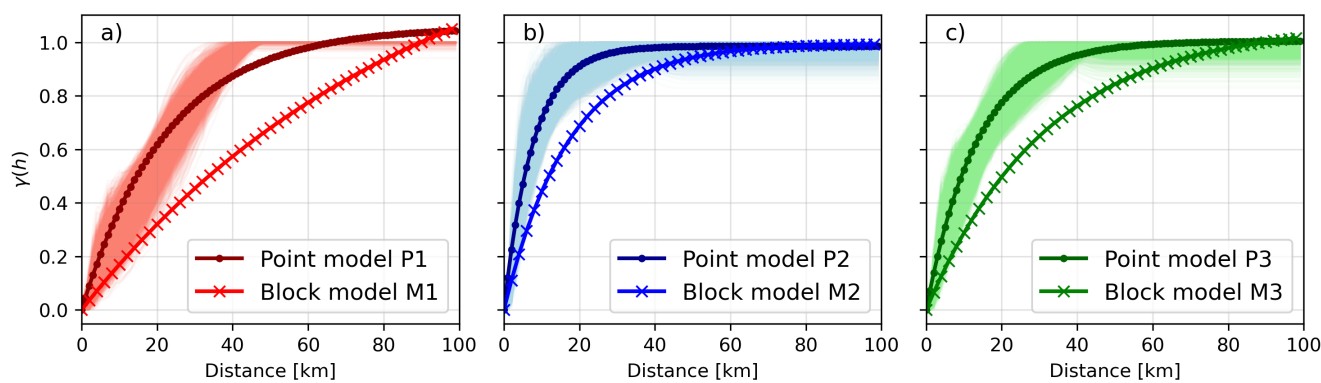

**Figure 8.** The calculated and clustered hourly experimental variograms using $DWD_{point}$ are shown in panels (a), (b), and (c), respectively. The theoretical models fitted to the average experimental variogram model of each cluster group (using $DWD_{point}$.) are denoted by Point P1, Point P2, and Point P3. These are displayed by the dark red, blue, and green curves. Similarly, the fitted theoretical models to the average cluster variogram using EURO-CORDEX RCP8.5 are denoted by Model M1 (light red curve in panel a), Model M2 (light blue curve in panel b), and Model M3 (light green curve in panel c), respectively.

For each time step in the RCP8.5 data, the corresponding variogram cluster was identified and the corresponding point scale variogram was used to rescale the spatial model. Without rescaling the grid cell variogram to the point variogram, the downscaled fields would be much smoother than the actual 1 km fields.

After correcting the spatial dependence structure and the bias in the CDF of the future RCP data, conditional realizations (conditioned on the model areal averages) at a finer spatial scale of 1 km (the weather radar grid) were generated using random

mixing. Hence, the downscaling part. Note that this section corresponds to part (c) in the flowchart shown in Figure 2. This process incorporates the climate signal from the projections and provides a dataset that can be used to derive future ADDF curves for the study area. To account for the uncertainty acquired by the simulation approach, 50 downscaled and equally probable time series for every pixel in the simulation domain were generated.

The step-by-step procedure to downscale the EURO-CORDEX RCP8.5 data for the period 2005-2099 for a given Area-DDF

location using random mixing is described below:

1. Create a buffer enclosing the ADDF largest area (1024 $km^2$).

2. Find all EURO-CORDEX grid cells falling within the buffer (conditional values).

3. Find all radar pixels falling within the buffer (simulation domain).





4. For every hour in the projected RCP8.5 data with precipitation > 1 mm, read corrected values.

5. Fit a non-parametric marginal distribution using Kernel density estimate with Gaussian kernel.

6. Derive the CDF and its inverse ($invcdf$) by optimizing the kernel width.

7. Transform the observations to standard normal space using the fitted CDF.

8. Fit an exponential covariance model.

9. Scale the model to match the reference point model.

10. Run conditional simulations (50 simulations)

11. Back-transform to original data space using $invcdf$.

12. Repeat for the next time steps.

An example of the simulation domain for one ADDF location is shown in panel (a) of Figure 9. The domain shown in red has a total area of 1024 $km^2$. Within this domain, the areas of 1, 16, 36, 100, 256, 576, and 1024 $km^2$ were considered for
deriving the ADDF curves. These are derived as DDF curves but by using a time series of the average of all pixels within each area. For example, for the area size of 1024 $km^2$, for every time step the average of the pixels enclosed by this area (pixels in red) was calculated. Using the same procedure for DDF curves (described in section 3.3), the ADDF curve for this area was derived. In panel (a), the gray pixels represent the complete simulation domain and the blue dots are the center of the EURO-CORDEX grid cells. The precipitation values at these grid cells were used as equality constraints in the downscaling
part. To showcase the influence of the variogram scaling procedure, two different spatial dependence models were used, model M2 (derived from EURO-CORDEX data) and model P2 (derived from DWD$_{point}$). Both can be seen in panel (b) of Figure 8. Because the simulations using random mixing are stochastic, it is difficult to compare them, hence, for comparison purposes, the average of 50 simulations was considered. Therefore, 50 realizations were generated using both models and the average fields were compared. Panels (b) and (c) in Figure 9 display the average fields using model M2 and model P2, respectively.
Note that the main difference in the spatial models is their range and how they approach the asymptotic limit. In panel (b), the larger range in model M2 is reflected by larger connected high (or low) rainfall grid cells and less variability in the final field, noted by the transition between wet (or high) and dry (or low) rainfall areas. The field in panel (c) based on model P2 has more variability and the smaller range is reflected by more discontinuities between the high and low values. Note that both average fields were conditioned on the same recorrelated and double-QQ corrected RCP8.5 data. Hence, to account for this difference
in the sub-scale downscaled fields, the grid cell variograms needed to be rescaled to the point scale.





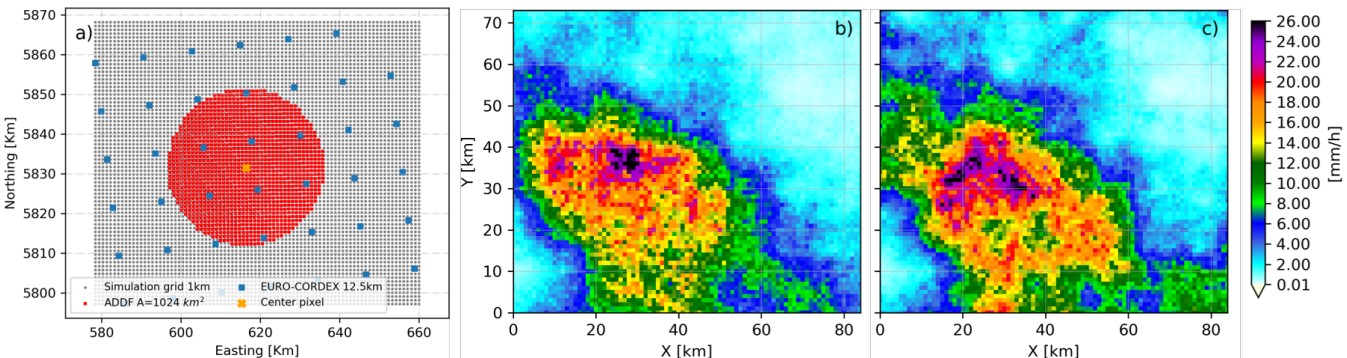

**Figure 9.** Panel (a) shows the 1-kilometre simulation grid domain (in gray) and the locations of the centre of the EURO-CORDEX grid cells (blue dots) within the ADDF largest area of 1024 $km^2$ (shown in red). Panels (b) and (c) display the average of 50 simulated fields using random mixing based on the RCP8.5 data (the blue dots in panel (a)), using the original grid cell variogram (Model M2), and the rescaled variogram (Point P2), respectively. Note that the field in panel (c) depicts larger variability.

## 4 Results

### 4.1 ADDF curves for future scenarios

#### 4.1.1 ADDF curves for small spatial scales

The downscaled fields are used to calculate the ADDF for selected regions/pixels in the study area. The aim is to derive areal extremes for future periods, particularly from the RCP8.5 data. The realizations generated by the conditional simulations are all equally probable but limited to spatial constraints and not temporally correlated (e.g., advection is not included). Therefore, for each time step with precipitation> 1 mm, 50 simulations were generated across the simulation domain for each duration separately. This was essential, otherwise, the fields would not be space-time continuous. For example, on the hourly scale, each realization for each time step will most likely be different from the realization of the next time step, despite being equally probable and statistically correct. Aggregating these fields will lead to an incorrect representation of the areal rainfall. A simple but computationally intensive solution was to aggregate the hourly corrected data for each required duration and rerun the simulations again. A different possible solution would have been to use the generated fields for time step $i$ as unconditional fields (instead of random fields) for time step $i+1$. This would have required modifying the simulation algorithm to include time as a third dimension. However, since the focus is on areal statistics, especially annual maxima, and not on event reconstruction, the first solution was seen as adequate enough for this scope. Moreover, an alternative stochastic simulation approach could have been tested. For example, Papalexiou et al. (2021) and Bárdossy and Hörning (2023) present frameworks for simulating space-time rainfall fields with characteristics such as velocity field, advection, anisotropy, and a flexible dependence structure.

After the simulations were completed, the ADDF curves for the area size of 1 $km^2$ were calculated from the generated time series for four different time periods, 2005-2025, 2026-2045, 2046-2064, and 2065-2099. For comparison purposes, the





simulations were performed using the raw data, the double-QQ data, and the recorrected and double-QQ corrected RCP8.5 data. The ADDF curve from RADOLAN data for the period 2005-2020 was calculated as reference data. Figure 10 shows an example of the ADDF values for the center pixel in the ADDF location (seen in panel (a) of Figure 9) for the return period of 5 years and 2 different time periods. Panel (a) for the period 2005-2025 and panel (b) for the period 2065-2099. Each boxplot for every duration consists of 50 simulations. The ADDF values for the different durations from the raw, the double-QQ corrected,

and the double-QQ and recorrelated RCP8.5 data are shown in the blue, orange, and green boxes, respectively. The blue crosses, the orange crosses, and the green dots represent the outliers in the raw and corrected RCP8.5 data, respectively. As reference data, the ADDF curve from RADOLAN data for the period 2005-2020 was calculated and is displayed in the red dashed line (and crosses).

    Compared to the RADOLAN data, the raw RCP8.5 values for both periods show an overestimation of the maxima over

all durations. The final corrected values (green boxes), however, fall within the range of the RADOLAN data and show a good agreement for the period 2005-2025. Additionally, the boxplots of the raw data indicate a larger range compared to the corrected data. For example, the estimated rainfall depth from raw data for the duration of 720 min (12 hours) varies between 40 and 100 mm with a mean value of 60 mm. However, in the corrected data, the range is between 25 and 40 mm. This indicates that the raw data has greater spread and variability, which makes it difficult to use a trustworthy uncertainty interval. Using the

double-QQ correction alone improved the results compared to the RCP8.5 raw data but was still far from the RADOLAN data. This also indicates the need for correction of the spatial dependence structure.

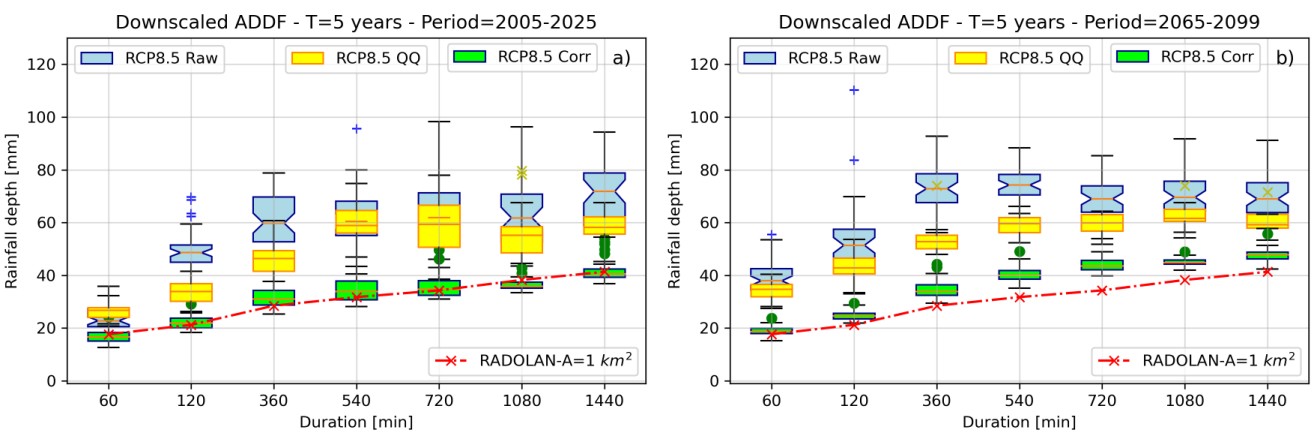

**Figure 10.** Derived ADDF curves (A=1 $km^2$) from RCP8.5 data before and after data correction for the ADDF center pixel for two different periods and a return period of 5 years. In panel (a) for the period 2005-2025 and panel (b) for the period 2065-2099. For every duration (x-axis), 50 simulations were generated and summarized in the boxplots. In both panels, the blue boxes refer to the raw RCP8.5 data, the yellow boxes to the double-QQ corrected data (without recorrelation), and the green boxes to the recorrelated and double-QQ corrected RCP8.5 data. The rainfall depth values derived from the RADOLAN data for the period 2005-2020 are displayed by the red crosses (or red curve).





In all data sets, an increase in the expected rainfall depth from the first to the second period is clearly visible. Though with different magnitudes. Panel (b) of Figure 10 shows the ADDF for the period 2065-2099 and indicates an increase in the expected rainfall depth for all durations and a return period of 5 years. However, the increase is not homogeneous across all

durations and varies accordingly.

### 4.1.2   ADDF curves for larger spatial scales

From the downscaled fields, the ADDF curves were derived from the raw, the double-QQ, and the recorrelated and double-QQ corrected fields and compared to the RADOLAN values. The results are displayed in Figure 11. The purpose of this analysis is to showcase if the pixel (point) and areal extremes show similar behaviour. The results of this approach are shown in panels

(a) and (b) of Figure 11. In panel (a) for the period 2005-2025 and in panel (b) for the period 2065-2099. The ADDF curve for the area size of A=1024 $km^2$ and return period of 5 years is shown and compared to the RADOLAN values. Compared to the RADOLAN values, the ADDF values from the raw and the double-QQ corrected data show an overestimation of the ADDF curves for all durations and both temporal periods. Similar results were noted for smaller area sizes (e.g., 16, 36, 100, 256, and 576 $km^2$). The final corrected data are in compliance with the RADOLAN results and indicate in panel (b) an increase in the

areal rainfall depth for all durations. Moreover, the raw data (boxes in blue) show a large uncertainty interval. The corrected data indicate, however, a smaller uncertainty interval across all durations. In addition, the uncertainty interval for larger areas is less than for smaller areas. In other words, the pixel ADDF curve shows the largest variations while the ADDF curve for A=1024 $km^2$ shows the smallest.

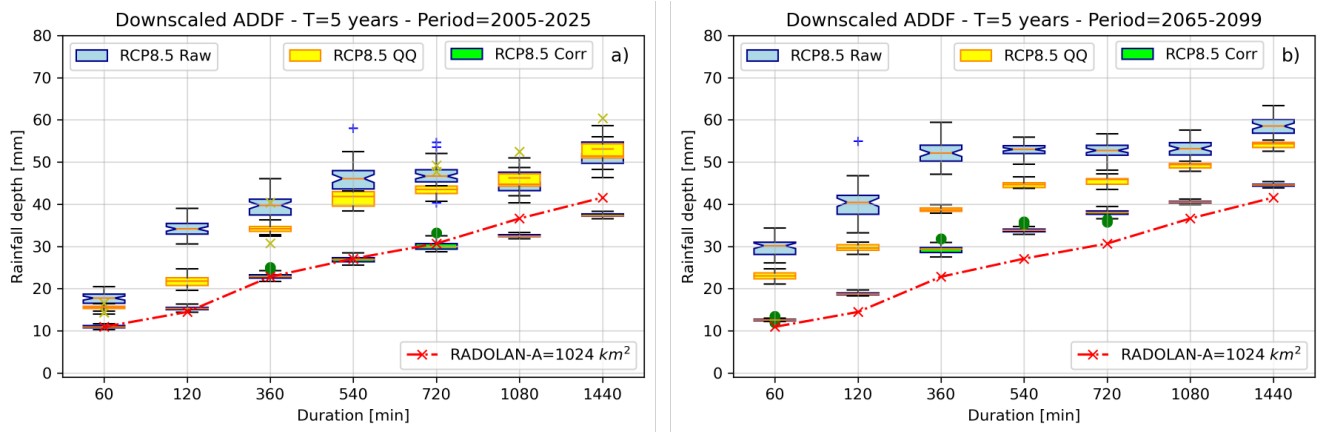

**Figure 11.** Estimated rainfall depth and ADDF curve from RCP8.5 data before and after data correction for the ADDF area of 1024 $km^2$ for a return period of 5 years. For every duration (x-axis), 50 simulations were generated and summarized in the boxplots. Panel (a) shows the results for the period 2005-2025. In panel (b), the ADDF curve for the period 2065-2099 is displayed. In both panels, the blue boxes refer to the raw RCP8.5 data, the yellow boxes to the double-QQ corrected data (without recorrelation), and the green boxes to the recorrelated and double-QQ corrected RCP8.5 data. The rainfall depth values derived from the RADOLAN data for the period 2005-2020 are displayed by the red crosses (or red curve).





Looking at the variation in the estimated rainfall depth across the different spatial scales (the ADDF areas), the latter was
found to be dependent on the duration. For example, the change between the ADDF values of the small areas (1 or 16 $km^2$)
compared to the larger scale (576 or 1024 $km^2$) is different on the hourly scale than on the 6-hour scale (for example). The
change between spatial scales decreases with increasing duration. Namely, a larger area reacts similarly to a smaller one for
long durations. To illustrate this, the 1 hour and 6 hours duration were chosen as an example. For the hourly scale, the difference
between the average estimated rainfall depth for the small area of 16 $km^2$ and the largest area of 1024 $km^2$ is around 25%
while for the 6-hour duration of 5%. Panel (b) of Figure 13 shows for two durations the estimated rainfall depth for each area
size separately. The data are derived from the final corrected data and for the time period 2065-2099. The green boxplots show
the hourly values and the orange boxplots the 18 hours values. The RADOLAN values are displayed by the red dots.

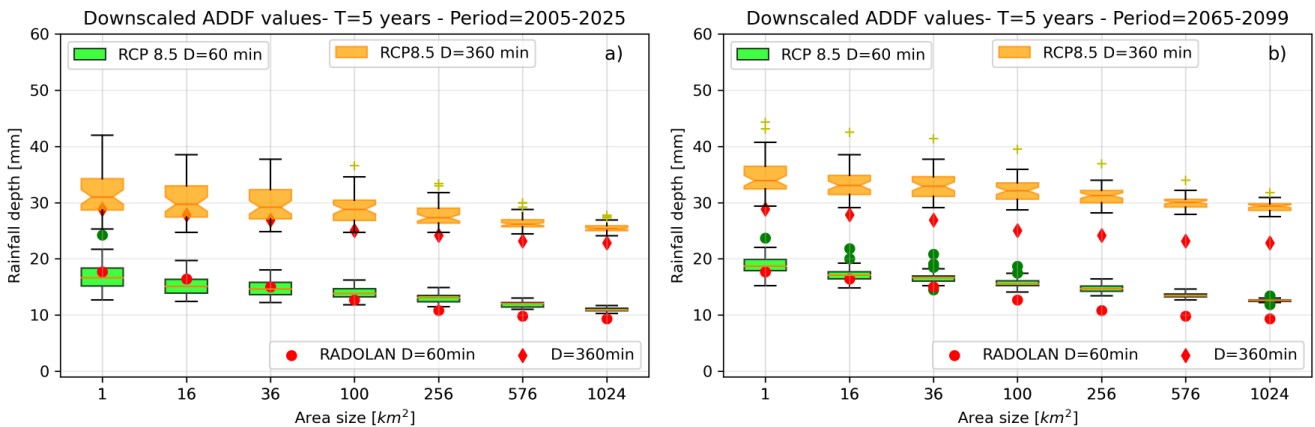

**Figure 12.** Estimated rainfall depth from RCP8.5 data after data correction for different area sizes, two selected durations (1 hour and 6
hours) and a return period of 5 years. For every duration, 50 simulations were generated and summarized in the boxplots. Panel (a) shows
the results for the period 2005-2025 and panel (b) for the period 2065-2099. The rainfall depth values derived from the RADOLAN data for
every duration and for the period 2005-2020 are displayed by the red dots. The latter are constant for the two time periods.

In Table 2, the average percentage of increase in the expected rainfall depth between the two time periods and for the different
durations and area sizes is shown. The percentage was calculated as the ratio between the mean estimated rainfall depth for
the two time periods 2005-2025 and 2065-2099 for the corresponding area size and duration. The results in Table 2 indicate
that the impact of climate change on the different spatial scales is duration dependent. For instance, on the hourly scale, the
percentage of increase is bigger for the larger areas than for the smaller areas, namely 15% for the area size of 1024 $km^2$ and
11% for the area size of 1 $km^2$. This difference is minimal for durations longer than 8 hours. For instance, on the 18-hour
duration, all area sizes have a similar percentage of increase ($\approx 23\%$). On the other hand, the percentage of increase changes
with the duration for a fixed area size. For example, for the area size of 100 $km^2$, the increase for the hourly duration is 12%
while for the daily duration, it is 28%. One would expect a monotonic increase with the area size and duration. This is mostly
the case, except for some area sizes and duration (for example, for the duration of 6 hours, the area size of 256 $km^2$ shows a





slightly larger average increase than the area of 576 $km^2$). Though, however, the change in the expected rainfall depth between the period 2005-2025 and the period 2065-2099 varies with the area size and duration. This indicates that applying a constant increase factor is not adequate and could lead to an over-/underestimation of the future areal extremes.

**Table 2.** Average percentage of increase in the expected rainfall depth between the periods 2005-2025 and 2065-2099 for the different durations and area sizes. The values correspond to the ratio between the average expected rainfall depth value for the two different time periods.

|  |  | Duration [min] | | | | | | |
|---|---|---|---|---|---|---|---|---|
|  |  | **60** | **120** | **360** | **540** | **720** | **1080** | **1440** |
| | **1** | 11.23 | 12.42 | 18.27 | 21.21 | 21.83 | 22.58 | 28.71 |
| | **16** | 12.31 | 12.49 | 20.3 | 22.54 | 24.2 | 22.51 | 28.64 |
| | **36** | 12.67 | 13.18 | 20.6 | 21.85 | 24.35 | 22.56 | 28.69 |
| Area size [$km^2$] | **100** | 12.75 | 15.4 | 20.65 | 21.62 | 24.98 | 22.83 | 28.99 |
| | **256** | 13.56 | 17.32 | 19.3 | 21.95 | 24.77 | 23.06 | 29.24 |
| | **576** | 13.77 | 20.1 | 18.49 | 22.19 | 25.2 | 23.14 | 29.29 |
| | **1024** | 14.88 | 22.81 | 20.27 | 22.55 | 25.81 | 23.17 | 29.32 |

## 4.2   Area reduction factor values for future scenarios

A final point to mention is that the ARF values can be calculated as the ratio between large and smaller ADDF curves. An example of this can be seen in Figure 13. Panel (a) shows the ARF calculated as the ratio between the estimated rainfall depth of 1024 $km^2$ and 1 $km^2$ area sizes for the period 2005-2025, using the raw data (blue boxes) and the final corrected data (green boxes). The x-axis denotes the duration (from hourly to daily) and the y-axis is the ARF value (usually between 0 and 1). The ARF derived from the RADOLAN values are shown by the red dots. The results in panel (a) indicate that the raw data underestimate the ARF, especially for longer durations. The corrected data show a better agreement with the RADOLAN data. Note that the ARF is traditionally used to transfer the DDF curves calculated from the rain gauge data to the catchment or areal scale. For instance, the RADOLAN data indicate that for the hourly duration the expected rainfall depth for the area size of 1024 $km^2$ is 62% of the center pixel rainfall depth. The corrected data indicate an average ARF of 0.64 (64%). As the duration increases, the ARF approaches the value of 1, indicating that for long duration small and large spatial scales behave similarly.

In panel (b) of Figure 13, the ARF values were calculated using the final corrected RCP8.5 data for two different temporal periods and for a return period of 5 years. For the period 2005-2025, the ARF values are presented by the green boxes. The ARF values for the period 2065-2099 are shown in the gray boxes. The results indicate that the future period has larger ARF values for short durations and similar for longer durations. This indicates that the partition of large to small spatial scale areal rainfall is changing for future periods. Namely, the increase for the large areas (1024 $km^2$) is greater than that of the smaller scale (1 $km^2$ ) rainfall depth. Note that similar results were noted for other areas.





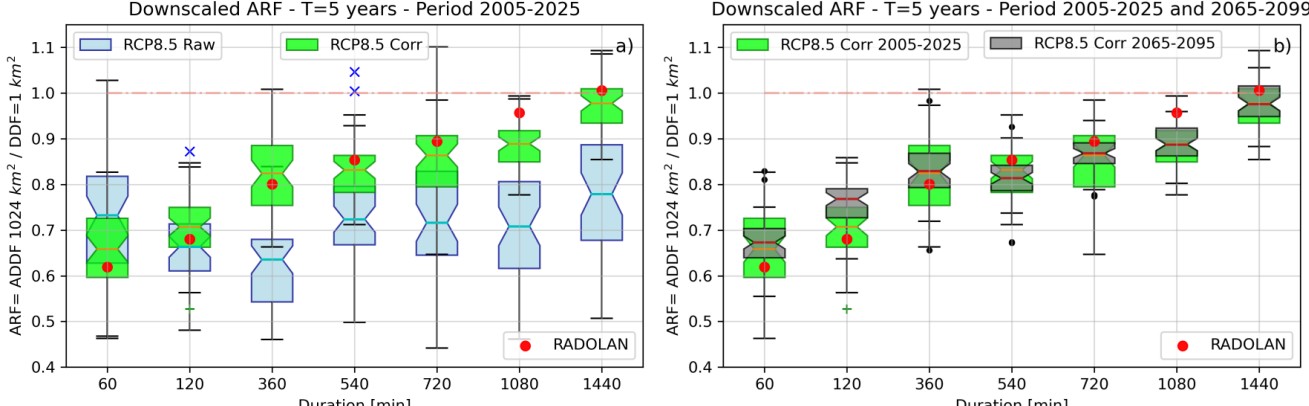

**Figure 13.** Panel (a) shows the estimated area-reduction-factor (ARF) from RCP8.5 data before and after data correction for a return period of 5 years and the time period of 2005-2025. The results are for the area sizes of 1 and 1024 $km^2$. For every duration (x-axis), 50 simulations were generated and summarized in the boxplots. In panel a), the blue boxes present the ARF from the raw data and the green boxes refer to the RCP8.5 corrected data. In panel (b), the ARF is derived from the corrected data from the time periods of 2005-2025 (green boxes) and period 2065-2099 (gray boxes). The ARF values derived from the RADOLAN data for the period 2005-2020 are displayed by the red dots in both panels.

### 4.3 Discussion

The final results indicate that the proposed methodology succeeded in obtaining reliable areal extremes from regional climate
models over several durations and spatial scales. Using the RCP8.5 data from the model without any correction showed an
overestimation of the derived ADDF curves across all temporal durations and considered periods. The overestimation was
evident in comparison to the values derived from RADOLAN-RW for the period 2005-2020. The double-QQ method corrected
the bias in the marginal distribution function but was not sufficient to achieve values in the range of the RADOLAN-RW data.
The recorrelation procedure, namely the correction of the spatial dependence structure, was needed. To handle the large number
of 0 mm values, the recorrelation had to be applied to the Gaussian transformed indicator correlation time series. This not only
improved the pair-wise Pearson correlation but also the rank correlation (Spearman correlation). Eventually, the combination
of the recorrelation method and the double-QQ transformation was needed. In fact, since the recorrelation procedure is based
on indicator correlations, applying it before or after the double-QQ mapping makes no difference. The indicator correlation is
not sensitive to quantile mapping. For downscaling to the finer spatial scale, the corrected data were used as conditional values
within a stochastic downscaling algorithm. This gave an uncertainty interval for each ADDF curve. However, as part of the
downscaling, the spatial dependence model needed to be rescaled to achieve larger small-scale variability. The final derived
ADDF curves were in the same range as the RADOLAN values for the period 2005-2020. The procedure used consisted of
many steps, but was necessary to obtain reliable results. After the correction and downscaling, the signal in the RCP projections
was not lost, the increase in areal precipitation values between the first period (2005-2025) and the last period (2065-2099)
was noted before and after the correction. However, the magnitude of this increase was smaller in the corrected data. The ARF





values were calculated from the ADDF curves and, after correction, showed a better agreement with the reference values. They show an increase in the values between the two time periods, especially for short durations. This refers to a change in the relationship between small and large-scale precipitation events.

The increase in the expected rainfall depth was obtained for small and large spatial scales (1 to 1024 $km^2$) and for short and
long durations. However, this increase was not constant but proportional to the increase in area size and duration. For shorter durations, the percentage of increase varied between 10 and 20%. For longer durations (above 8 hours) between 20 and 30%. The results are consistent with other studies that examined the influence of climate change on precipitation extremes. However, the results in this work provide insight into possible future values for areal extremes at various spatial scales (and not just the point scale). Furthermore, the effect of the duration was stronger than the effect of area size. If the area size is constant,
the percentage of increase with the duration is greater than if the duration is constant, and the area size increases. The largest increase occurs for the daily duration and largest area size. The analysis of future areal extremes is essential for adaptation strategies. For practical purposes such as the design of urban drainage systems, rainfall depth values associated with relatively short return periods of 5 years are required. Moreover, the uncertainty interval obtained from the downscaling scheme provides an ensemble of values needed for risk analysis.

The ADDF curves were derived using the DWA-A 532 procedure with the Gumbel distribution. However, there are alternative ways to calculate the DDF curves. For instance, Koutsoyiannis et al. (1998) and Fischer and Schumann (2018) offer alternative frameworks. Subsequent derived ADDF values will change, and the corresponding results may as well. Furthermore, the results shown in this work are only valid for relatively small areas (mesoscale catchment size) of up to 1000 $km^2$. For larger areas, the areal mean precipitation is less of interest, while the areal distribution becomes more relevant. The corre-
lation structure was assumed to be homogeneous and not changing with time. In other terms, the spatial dependence structure is assumed to not be influenced by climate change. An aspect that is already present in the raw EURO-CORDEX data (see Figure A1). This assumption may not be valid. However, the presented approach using the indicator correlation is more robust, although it assumes a multi-normal dependence structure. Which may not always be valid. The final results show close agreement with the reference RADOLAN values, although the latter may underestimate the areal extremes.

**5   Conclusion**

Investigating the changes in the statistical properties of climatic variables, such as rainfall due to a changing climate, is essential for coping and preparing for future periods. Regional climate models provide useful information about climatic data for historical and future scenarios. These have been generated based on increasing emission levels, hence, changing the physical, energetic, and thermodynamic balance between the atmospheric components. The outcome of RCM data is in general too
coarse for local analysis and often spatial or/and temporal downscaling is applied. For having reasonably downscaled values, the original RCM data should be inspected and eventually corrected. Here, two major aspects are crucial. The first is related to the spatial dependence structure. This influences the distribution of areal rainfall, and a false structure alters any subsequent results. The second is related to the presence of a bias in the data of future periods. A bias in any direction (over-/underestimation)




of the marginal distribution function affects the temporal structure and the quality of the data. If the model distribution differs largely from the observed one, it would be as if it was not a realization of the same process. Hence, the first and second parts of this manuscript were related to correcting the spatial dependence structure and the marginal distribution function, respectively. Both of these steps were undertaken on the same spatial and temporal scale as the model data. This required upscaling the reference data to the model scale. The corrected data can now be integrated into a downscaling scheme. In this part, a probabilistic scheme involving conditional simulations using random mixing was applied. For each time step and duration, several realizations on the 1-km scale were generated and used for analyzing areal extremes. The spatial dependence model derived from the grid cell scale was rescaled to the point scale to have higher variability in the simulated fields. From the final fields, ADDF curves along ARF values were derived for different temporal periods and spatial scales, and compared to the RADOLAN derived values. The results indicate for the current period (2005-2025) a good agreement with the RADOLAN values and showcase an increase in the areal extremes over all durations and temporal scales for the period 2065-2099. The final results show realistic ADDF curves and ARF values that can be used for impact analysis.

Main messages behind this manuscript:

1. Using regional climate model data directly leads to an overestimation of the areal extremes.

2. Correcting the spatial dependence structure and the magnitudes of the data improves the usability of the data.

3. Spatial downscaling using random mixing is possible and offers uncertainty intervals.

4. The signal in the RCP is not lost and is present in the future areal extremes.

5. The corrected data of the RCP8.5 scenario indicate an increase in the rainfall depth over all temporal and spatial scales.

The study could be further extended and applied to several GCM-RCM products. This will provide an ensemble of possible uncertainty scenarios for future projections of DDF and ADDF curves. Although the procedure would be similar for the different GMC-RCM combinations, the results might differ. The procedure could be also further evaluated for other geographical locations. Moreover, instead of working with ADDF areas, a selected catchment could have been used. The resulting future areal precipitation extremes could be integrated into a hydrological model to assess the impact on the discharge values.

*Data availability.* The precipitation rain gauge and RADOLAN data were obtained from the Climate Data Center of the Deutscher Wetterdienst (https://opendata.dwd.de/climate_environment/CDC). The EURO-CORDEX 11 data were made available by the ClimXtreme Central Evaluation System framework (Kadow et al., 2021).

*Code availability.* The corresponding code is available upon request from the contact author. The random mixing simulation code is available within the RMWSPy Python package (Hörning and Haese, 2021).



*Author contributions.* AEH developed and implemented the algorithm for the study area. JS assisted in the downscaling section. AB designed and supervised the study. All authors contributed to the writing, reviewing, and editing of the manuscript.

*Competing interests.* At least one of the (co-)authors is a member of the editorial board Hydrology and Earth System Sciences.

590 *Acknowledgements.* This study is part of the project B 2.7 "STEEP - Space-time statistics of extreme precipitation" (Grant No. 01LP1902P) of the "ClimXtreme" project funded by the German Ministry of Education and Research (Bundesministerium für Bildung und Forschung, BMBF). The authors thank the German Weather Service DWD for providing the precipitation and weather radar data and the EURO-CORDEX community for providing the climate model data. We thank Masoud Mehrvand for assisting in writing the code for transforming the indicator to Gaussian correlation. Moreover, we acknowledge all developers of different Python core libraries (e.g., numpy, pandas, mat-
595 plotlib, cython, scipy) for providing open-source code. The authors thank the University of Stuttgart for funding this open-access publication.





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





**Appendix A: Additional figures:**

**A1    Rank correlation values for the EURO-CORDEX historical and future data.**

Figure A1 shows the pair-wise grid cell rank correlation values for the EURO-CORDEX historical and future data. Panel (a)
740    shows the results for the winter period and panel (b) for the summer period. For the latter, the correlation values seem to be
highly similar, with few differences. However, for the winter period, the historical data show a quicker drop in the correlation
values than the future data. The agreement between the historical and future structure is very high and indicates a stable
dependence structure between the two periods.

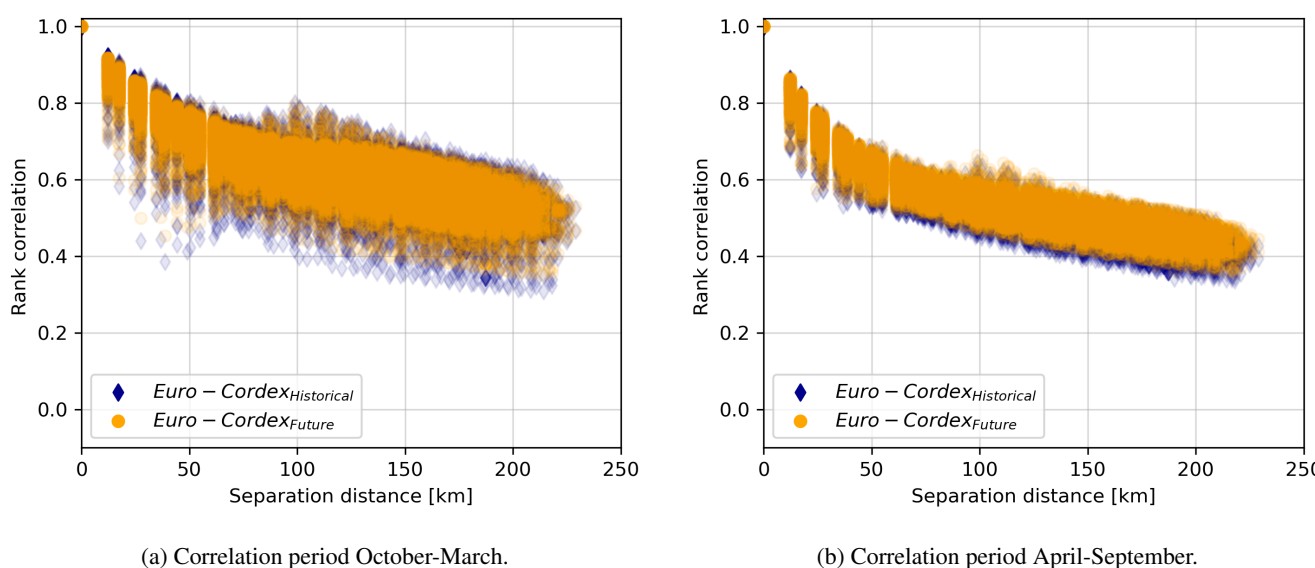

(a) Correlation period October-March.

(b) Correlation period April-September.

**Figure A1.** Rank correlation dependence structure for historical (blue dots) and future (orange dots) EURO-CORDEX data for winter (a)
and summer (b) periods.

**A1    Relation between the indicator and Gaussian correlation values.**

745    Figure A2 presents the relation between the indicator and Gaussian correlation values for different thresholds $\alpha$. Note that
the curves are only shown for the positive correlation domain [0, 1]. Each curve is obtained by solving the relation defined in
equation 4 for the corresponding $\alpha$.





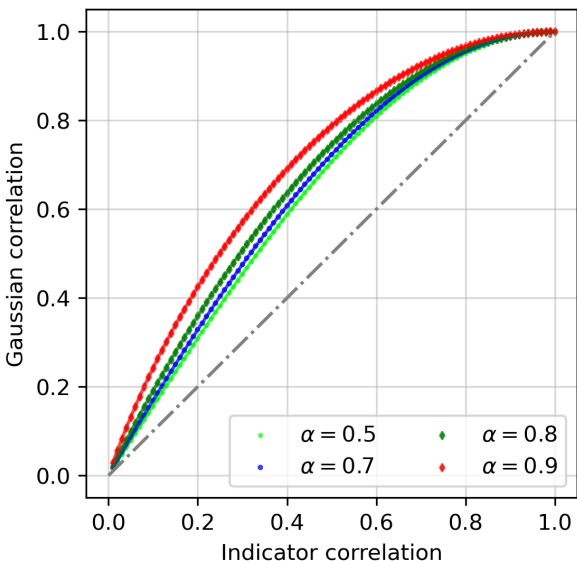

**Figure A2.** The relation between the indicator and Gaussian correlation used to transform the indicator correlation matrices to Gaussian correlation matrices. Each curve corresponds to a different probability threshold $\alpha$. Note that only positive correlation values are shown.

## A2  Theoretical variograms parameters

Table A1 shows the estimated parameters of the three theoretical exponential variogram models. These were fitted to the average empirical variogram of every cluster (shown in Figure. 8) using the $\text{DWD}_{point}$ and the EURO-CORDEX data.

**Table A1.** Parameters of the fitted experimental variograms using $\text{DWD}_{point}$ and the EURO-CORDEX grid cell data

|  | Sill | Range [km] |
|---|---|---|
| **Point model 1 (P1)** | 1.06 | 22.76 |
| Grid cell model 1 (M1) | 1.55 | 86.45 |
| **Point model 2 (P2)** | 0.99 | 7.33 |
| Grid cell model 2 (M2) | 1 | 17.05 |
| **Point model 3 (P3)** | 1.01 | 13.57 |
| Grid cell model 3 (M3) | 1.06 | 31.82 |