# Peer review of "Probabilistic downscaling of EURO-CORDEX precipitation data for the assessment of future areal precipitation extremes for hourly to daily durations"

_Hydrology and Earth System Sciences, 2023_

## Author Comment (AC2)

**Response to reviewer #2**

We thank the reviewer for his comments and for taking the time to review our manuscript. Our answers to the two specific comments are show in blue.

*I'd like to see the criteria used by the authors to choose MPI-M-MPI-LR-GERICS-REMO2015-v1 as a single model to assess the climate change impact instead of using an ensemble from EURO-CORDEX.*

We agree that using a large number of different climate model outputs can show the uncertainty of the predictions with regard to the input uncertainty. However, the goal of our paper is to present a reasonable chain of methods together with their importance for downscaling. As stated in the paper the choice of the model was aligned with the data availability (at that time) within the ClimXtreme Central Evaluation System framework. In principle any other model data could be used as well.

*Also, I would like to see 1-2 sentences regarding your experience with extrapolation issue in Line 265-270 in case the modelled precipitation is extremely larger than the observed extreme.*

It is true that the exponential function can lead in some cases to an overestimation of the precipitation extremes. For those cases, the extrapolation can be restricted by a linear component following the procedure described in Yan et al. (2020). This will be added to the revised manuscript.

$$F(z) \ = \ \min\left(1 - e^{-\lambda z}, \frac{U_m - U_{m-1}}{Z_m - Z_{m-1}} + U_m\right)$$

With $(Z_m, U_m)$, $(Z_{m-1}, U_{m-1})$ defining the pairs of the largest two precipitation observations and corresponding quantiles.

Reference:

Yan, J., Bárdossy, A., Hörning, S., & Tao, T. (2020). Conditional simulation of surface rainfall fields using modified phase annealing. Hydrology and Earth System Sciences, 24(5), 2287-2301.

Minor comments

All these remarks will be accounted for in the updated manuscript.

---

## Author Response (AR1)

We thank the Editor for her comments and suggestions. In the following letter, we would like to summarize the changes to the new manuscript.

Our aim was to incorporate all aspects mentioned by the Editor. Concerning the major comments, we incorporated several comments from reviewer 1 in the discussion and conclusion. There were indeed relevant aspects to be mentioned. These were divided between the discussion and conclusion as seen adequate. To showcase the effect of the extrapolation we added a section in the appendix explaining the procedure in further details. We created summary statistics of the extrapolated values (hourly resolution) and compared them to the original and to the reference values. We hope that this will provide additional insights to interested readers. To avoid increasing the length of the main manuscript we decided to add this to the appendix and referenced it in the main text. We rewrote the methods part and detailed the recorrelation procedure to provide more clarity and completeness. Moreover, we replaced Figure 6 in the manuscript by a scatter plot of the correlations before and after recorrelation. We think this is better to display the effect of the recorrelation. We changed Figure 9 to make colorblind friendly. In addition, Figure 1 in the appendix was replaced with a new figure show chasing a scatter plot of the correlation values between the historical and future Euro-Cordex data. The message behind the figure remains the same, namely that the dependence structure does not change. All other minor comments were incorporated in the revised manuscript.

We thank you for handling our manuscript,

The authors

---

## Author Response (AR2)

We thank the Editor for all her suggestions. We tried to incorporate all of them in the revised manuscript. In addition, we checked all figures using the Coblis – Color Blindness Simulator and found no issues with them.

Thank you for handling our manuscript.

On behalf of all authors,

Abbas El hachem